# Reproducibility Study of
# CDUL: CLIP-Driven Unsupervised Learning for Multi-Label Image Classification

## Abstract

This report is a reproducibility study of the paper *CDUL: CLIP-Driven Unsupervised Learning for Multi-Label Image Classification (Abdelfattah et al., 2023)*. Our report makes the following contributions: (1) We provide a reproducible, well commented and open-sourced code implementation for the entire method specified in the original paper. (2) We try to verify the effectiveness of the novel aggregation strategy which uses the CLIP (Radford et al., 2021a) model to initialize the pseudo labels for the subsequent unsupervised multi-label image classification task. (3) We try to verify the effectiveness of the gradient-alignment training method specified in the original paper, which is used to update the network parameters and pseudo labels.

## 1 Introduction

Multi-label image classification represents a complex challenge within the field of computer vision, marked by the necessity to identify multiple objects or attributes within a single image. This task is further complicated by the scarcity of comprehensively annotated datasets, which are essential for training supervised learning models but are expensive and labor-intensive to produce. The reliance on extensive, accurately labeled data poses a significant bottleneck, limiting the applicability of advanced models in real-world scenarios where images often contain rich, diverse content. Unsupervised learning approaches offer a promising alternative, aiming to leverage existing unlabeled data effectively, thereby circumventing the need for manual annotation and potentially unlocking new capabilities in multi-label image classification.

To tackle this issue, Abdelfattah et al. (2023) propose CDUL (CLIP-Driven Unsupervised Learning), an unsupervised learning method for multi-label image classification that leverages the powerful capabilities of the CLIP (Contrastive Language-Image Pre-training) model (Radford et al., 2021b). The central idea behind CDUL is to exploit the rich semantic knowledge encoded within CLIP, which was pre-trained on a vast collection of image-text pairs, to generate high-quality pseudo labels for unlabeled images. These pseudo labels serve as a proxy for manual annotations, enabling the training of a multi-label classification model without the need for human-annotated data.

### 1.1 Background on the CDUL Method

The CDUL method comprises three main stages: initialization, training, and inference. In the initialization stage, the authors propose a novel approach to extend CLIP for multi-label predictions based on a global-local image-text similarity aggregation strategy. Specifically, they split each image into snippets and leverage CLIP to generate similarity vectors for the whole image (global) as well as for each individual snippet (local). These global and local similarity vectors are then combined through a similarity aggregator, resulting in a set of initial pseudo labels that capture the multi-label nature of the input image.

During the training stage, the authors introduce an optimization framework that utilizes the generated pseudo labels to train the parameters of a classification network. Crucially, they propose a gradient-alignment

method that recursively updates not only the network parameters but also the pseudo labels themselves. This iterative refinement process aims to minimize the loss function by aligning the predicted labels with the updated pseudo labels, effectively learning to predict multiple relevant labels for unseen images.

The key components of the CDUL method are as follows:

1. **Global Alignment Based on CLIP**: Given an input image $x$, CLIP's visual encoder $E_v$ maps it to an embedding vector $\mathbf{f}$. The relevant similarity score between $\mathbf{f}$ and the text embedding $\mathbf{w}_i$ for class $i$ is given by:

$$p_i^{glob} = \frac{\mathbf{f}^\top \mathbf{w}_i}{||\mathbf{f}|| \cdot ||\mathbf{w}_i||} \tag{1}$$

$$s_i^{glob} = \frac{\exp(p_i^{glob}/\tau)}{\sum_j \exp(p_j^{glob}/\tau)} \tag{2}$$

where $\tau$ is a temperature parameter learned by CLIP, and $s_i^{glob}$ is the normalized similarity score for class $i$ using a softmax function.

2. **CLIP-Driven Local Alignment**: To generate local alignments, the input image is split into $N$ snippets $r_j{}_{j=1}^N$. For each snippet $r_j$, the visual embedding $\mathbf{g}_j$ is extracted from CLIP's visual encoder: $E_v(r_j) = \mathbf{g}_j$. The cosine similarity scores between $\mathbf{g}_j$ and the text embedding $\mathbf{w}_i$ for class $i$ are computed as:

$$p_{j,i}^{loc} = \frac{\mathbf{g}j^\top \mathbf{w}_i}{||\mathbf{g}_j|| \cdot ||\mathbf{w}_i||} \tag{3}$$

$$s_{j,i}^{loc} = \frac{\exp(p_{j,i}^{loc}/\tau)}{\sum k \exp(p_{j,k}^{loc}/\tau)} \tag{4}$$

These local similarity scores are then aggregated into a local soft similarity vector $\mathbf{S}_j^{local}$ for each snippet.

3. **Global-Local Image-Text Similarity Aggregator**: The global and local similarity vectors are aggregated using a min-max strategy to form a unified local similarity vector $\mathbf{S}^{aggregate}$ for each image:

$$\gamma_i = \begin{cases} 1, & \text{if } \alpha_i \geq \zeta \\ 0, & \text{otherwise} \end{cases} \tag{5}$$

$$s_i^{ag} = \gamma_i \alpha_i + (1 - \gamma_i)\beta_i \tag{6}$$

where $\alpha_i = \max_j s_{j,i}^{loc}$, $\beta_i = \min_j s_{j,i}^{loc}$, and $\zeta$ is a threshold parameter.

The final similarity vector $\mathbf{S}^{final}$ is then computed as the average of the global and aggregated local similarity vectors:

$$\mathbf{S}^{final} = \frac{1}{2}(\mathbf{S}^{global} + \mathbf{S}^{aggregate}) \tag{7}$$

This $\mathbf{S}^{final}$ vector serves as the initial pseudo labels for the unobserved labels during the training stage.

4. **Gradient-Alignment Network Training**: During training, the pseudo labels $\mathbf{y}_u$ are initialized from $\mathbf{S}^{final}$. The network parameters are updated based on the Kullback-Leibler (KL) divergence loss between the predicted labels $\mathbf{y}_p$ and the pseudo labels $\mathbf{y}_u$. Subsequently, the latent parameters of the pseudo labels $\tilde{\mathbf{y}}_u$ are updated using the gradient of the loss function with respect to $\mathbf{y}_u$:

$$\tilde{\mathbf{y}}_u = \tilde{\mathbf{y}}_u - \psi(\mathbf{y}_u) \odot \nabla_{\mathbf{y}_u} L(\mathbf{Y}_u|\mathbf{Y}_p, \mathbf{X}) \tag{8}$$

where $\psi(\mathbf{y}_u)$ is a Gaussian distribution centered at 0.5, and $\odot$ denotes element-wise multiplication. This process alternates between updating the network parameters and the pseudo labels, aiming to minimize the total loss function $L(\mathbf{Y}_p, \mathbf{Y}_u|\mathbf{X})$.

During inference, only the trained classification network is used to predict the labels for new input images, without requiring any manual annotations.

## 2 Report Contributions

In this reproducibility study, we aim to verify the effectiveness of the proposed global-local image-text similarity aggregation strategy and the gradient-alignment training method. We provide an open-source implementation of the CDUL method and conduct experiments on the PASCAL VOC 2012 dataset to evaluate the validity of the authors' claims. Through a comprehensive analysis, we seek to assess the robustness and potential of this unsupervised approach for multi-label image classification tasks.

## 3 Scope of reproducibility

This report attempts to verify the following central claims of the original paper:

1. The effectiveness of the aggregation of global and local alignments generated by CLIP in forming pseudo labels for training an unsupervised classifier.

2. The effectiveness of the gradient-alignment training method, which recursively updates the network parameters and the pseudo labels, to update the quality of the initial pseudo labels.

To verify the claims made by the authors, it is necessary to conduct an independent reproducibility study on the datasets, methods and hyperparameters that the authors specify.

## 4 Reproducibility Methodology

Since there is no availability of a public codebase or a paper supplementary, we create a well commented codebase to the best of our understanding, for verifying the central claims of the paper. For verifying **claim 1**, we need to compute both the global similarity vectors and the local similarity vectors on the snippets of an image, for all images of the dataset. After computing the local similarity vectors, an aggregate vector is computed, which is then averaged with the global similarity vector to produce the initial pseudo labels for the dataset. If the claim holds, then the mean average precision (**mAP**) for the pseudo label vectors for a dataset split should be higher than the **mAP** of the pseudo labels when initialized using only the global similarity vectors obtained from CLIP. For verifying **claim 2**, we need to train a classifier on the training set by setting the targets of the dataset to the pseudo labels generated in the previous step. If the claim holds, then the predictions from the classifier should improve over training epochs along with the improvement in the quality of pseudo labels. This means that we should be able to see an increase in the **mAP** of the predictions and the pseudo labels over training epochs.

### 4.1 Model descriptions

The authors use two models in their overall method:

1. CLIP: The initialization stage uses the CLIP model with ResNet-50 (He et al., 2016) as the image encoder to generate similarity vectors for the global-local aggregation strategy to generate the pseudo labels for the unlabeled data. For encoding the text, a fixed prompt, "a photo of a [class]", where class denotes the class labels of a dataset, is used.

2. ResNet-101 (He et al., 2016): Once the pseudo labels are generated for the unlabeled training set using CLIP, a ResNet-101 classifier pre-trained on ImageNet (Deng et al., 2009) is used for training and later for inferencing.

Here it is worth mentioning that although the authors stressed on the fact that the whole approach is cost-effective during both training and inference phases, we differ from this view since generating the local alignment vectors for a snippet size of $3 \times 3$ requires a lot of inferencing through the CLIP model for all images of the dataset.

## 4.2 Datasets

The original paper conducts experiments on four datasets: PASCAL VOC 2012 (Everingham et al., 2012), PASCAL VOC 2007, MS-COCO (Lin et al., 2014) and NUSWIDE (Chua et al., July 8-10, 2009). Since generating the pseudo labels with a small snippet size is a compute and time intensive process, we only tried to verify the main claims on the PASCAL VOC 2012 dataset. This dataset contains 5717 training images and 5823 images in the official validation set, which is used as the test set. Labels from the dataset are in an `XML` format, where different objects found in the image are under the `object` field. An element from the `object` list further contains a boolean field named `occluded`, which specifies whether an object is occluded. Since the authors do not explicitly mention about ignoring occluded objects as a multi-label class, we do not make any such assumptions and include occluded objects as well, in the ground truth one-hot encoded multi-label vector.

## 4.3 Hyperparameters

For the first part, i.e generating the pseudo label vectors, we test their quality using the following snippet sizes: [64, 32, 16, 3]. Since the authors do not specify the value of the threshold parameter $\zeta$ from section 3.1.3 "Global-Local Image-Text Similarity Aggregator" of the original paper, we assign $\zeta$ a logical value of 0.5. We observed that changing this parameter to 0.3 or 0.7 did not affect the initial **mAP** of the pseudo labels. Since section 3.2 "Gradient-Alignment Network Training" of the original paper doesn't specify any pseudocode or epoch frequency at which the pseudo labels are being updated, we consider that as a hyperparameter and experiment over the values: [1, 10]. We set the value of $\sigma$ to 1 for the Gaussian distribution, since it isn't specified as well. The ResNet-101 classifier is trained end-to-end while updating the parameters of the backbone and classifier, with the original values of batch size = 8, learning rate = $10^{-5}$. For verifying the original claims, we set the epochs to 20 and pseudo label update frequency to 1. We train for 100 epochs when the pseudo label update frequency is set to 10.

## 4.4 Experimental setup and code

Since there is no public codebase available, as part of the main contribution of our report, we create a well commented codebase, referring to relevant equations, to the best of our understanding of the original paper. We use PyTorch as our deep learning framework. The entire codebase is configuration driven with the help of `Hydra` and `Makefile`. We provide well structured configuration files for all the experiments, which can be run using simple `make` commands, making all our experiments completely reproducible. We provide a `README.md` file detailing out the repository setup and reproducibility of our experiments. Since generating the aggregate vectors from section 3.1.3 "Global-Local Image-Text Similarity Aggregator" requires inferencing CLIP on several 3x3 snippets of all images from the dataset, we cache the generated global and aggregate vectors for future use in running multiple experiments for training the classifier. As part of our contribution, we provide the generated cache as well, since its computation can be time intensive, as detailed out in the computational requirements section.

## 4.5 Evaluation metrics

The mean average precision (**mAP**) across all the classes is used as the metric for evaluating the methods for the task of multi-label image classification. For an easy and robust implementation, we utilise the `MultilabelAveragePrecision` [1] metric from the `TorchMetrics` [2] library along with a `ClasswiseWrapper` [3] to evaluate the average precision (**AP**) per class.

## 4.6 Computational requirements

We conducted all our experiments on a single NVIDIA Tesla V100-SXM2-32GB GPU. If there are **N** images in the dataset (the train split) with an average image size of $\mathbf{a} \times \mathbf{a}$, then the total number of snippets formed

---

[1] `https://lightning.ai/docs/torchmetrics/stable/classification/average_precision.html`

[2] `https://lightning.ai/docs/torchmetrics/stable/`

[3] `https://lightning.ai/docs/torchmetrics/stable/wrappers/classwise_wrapper.html`

with a snippet size $\mathbf{k} \times \mathbf{k}$ is of the order $\mathcal{O}(N\frac{a^2}{k^2})$. Table 1 lists the approximate time taken to generate the global and aggregate vector caches for the PASCAL VOC 2012 dataset on the train split containing 5717 images.

Table 1: Approximate time for cache generation for different snippet sizes

| Cache | Approx. Time Taken |
|---|---|
| Global | 3min 49s |
| $64 \times 64$ Aggregate | 2hr 14min |
| $32 \times 32$ Aggregate | 10hr 50min |
| $16 \times 16$ Aggregate | 41hr 15min |
| $3 \times 3$ Aggregate | >30 days |

The values in table 1 show that computing the cache becomes extremely time intensive with a decrease in the snippet size. Once the cache has been generated, training the classifier is relatively less compute and time intensive. Our best run of 100 epochs with a pseudo label update frequency of 10, took 3hr 55min to run.

## 5 Results

### 5.1 Claim 1

The original paper directly uses a snippet size of $3 \times 3$ in generating the aggregate vectors, which in turn are used along with the global similarity vectors to initialize the pseudo labels for the unlabeled training dataset. Table 2 shows the **mAP** on the train split of PASCAL VOC 2012, using the final pseudo labels obtained using the "Global-Local Image-Text Similarity Aggregator" method from the original paper.

Table 2: **mAP (in %)** of the pseudo label vectors generated using different snippet sizes on the train split of the PASCAL VOC 2012 dataset. Here "Global" indicates that pseudo labels were generated directly from the similarity scores obtained from CLIP on the entire image (without any averaging with the aggregate vectors).

| Snippet Size | Ours | Table 3 of the original paper |
|---|---|---|
| Global | **85.9** | 85.3 |
| $64 \times 64$ | 84.62 | (Not computed) |
| $32 \times 32$ | 84.99 | (Not computed) |
| $16 \times 16$ | 85.19 | (Not computed) |
| $3 \times 3$ | 85.47 | 90.3 |

The **mAP** values computed by us for the global alignment vectors differ slightly (by **0.6%**) from the ones reported in the original paper. Although section 4.3 of the original paper suggests an increase in the quality of pseudo labels from the global counterparts to be nearly **+5%** (for the PASCAL VOC 2012 dataset) using the "Global-Local Image-Text Similarity Aggregator" strategy, we observe the contrary. For all snippet sizes in our experiments, the **mAP** is lower than the global counterpart. We observe a very small increase the **mAP** as the snippet size decreases from $64 \times 64$ to $3 \times 3$.

### 5.2 Claim 2

For verifying claim 2, we train a ResNet-101 classifier using ImageNet pre-trained weights on the generated pseudo labels from the previous step. In our case, since the quality of pseudo labels did not improve over the global counterpart using the "Global-Local Image-Text Similarity Aggregator" strategy, we directly use the global image-text similarity vectors ($S^{\text{global}}$) as the initial pseudo labels (denoted by $\mathscr{P}\ell$).

Table 3: Gradient-Alignment Network Training

| Epochs | $\mathscr{P}\ell$ Update Frequency (in epochs) | Train **mAP** (Ours) | $\mathscr{P}\ell$ **mAP** (Ours) | Val **mAP** (Ours) | Val **mAP** (Original) |
|---|---|---|---|---|---|
| 20 | 1 | 31.1 | 67.9 | 23.1 | 88.6 |
| 100 | 10 | 84.8 | **86.1** | 70.6 | - |

Table 3 shows the experiment using the original hyperparameters proposed by the authors and the additional one where we train for a longer duration of 100 epochs along with a pseudo label update frequency of 10 epochs (this basically means that the latent parameters of the pseudo labels are updated after 10 epochs instead of every epoch). We observe that the original hyperparameters lead to a poorly trained network (see Fig 1 top row), whereas increasing the training epochs and decreasing the pseudo label update frequency to once per 10 epochs leads to a much better val **mAP** (see Fig 1 bottom row). Updating pseudo labels every epoch causes their quality to decrease from an original **85.9 mAP** to 67.9 in 20 epochs. Decreasing the pseudo label update frequency to once per 10 epochs leads to a steady increase in the quality of pseudo labels over epochs. However, even with this strategy, the quality of pseudo labels only reaches an **mAP** of **86.1**, which is just a **0.2%** improvement over the initialized value. Although we observe the quality of pseudo labels to increase marginally, the overall validation **mAP** only reaches a value of 70.6%, which is far below the training counterpart and the value reported in the original paper.

## 6 Discussion

### 6.1 Verifying claims

**Claim 1 -** We were unable to verify claim 1 due to the time intensive computation of generating the aggregate vectors for a snippet size of $3 \times 3$. In all our ablation experiments with different snippet sizes ranging from relatively larger $64 \times 64$ ones to the smallest $16 \times 16$ ones, we found the **mAP** of the final pseudo label vectors to be lesser than their global counterparts. We however observed the **mAP** of the final pseudo label vectors to increase, as the snippet sizes decreased from $64 \times 64$ to $16 \times 16$.

**Claim 2 -** We were unable to reproduce claim 2 of the original paper completely using the specified hyperparameters. However, we did observe an improvement in the quality of pseudo labels across training epochs when decreasing the frequency of updating the latent parameters of the pseudo labels once every 10 epochs. Hence, we were only able to weakly support the "Gradient-Alignment Network Training" claim of improving the quality of pseudo labels during training.

### 6.2 What was easy

The ideas presented in the paper were relatively well structured, making it easy for us to implement them using PyTorch.

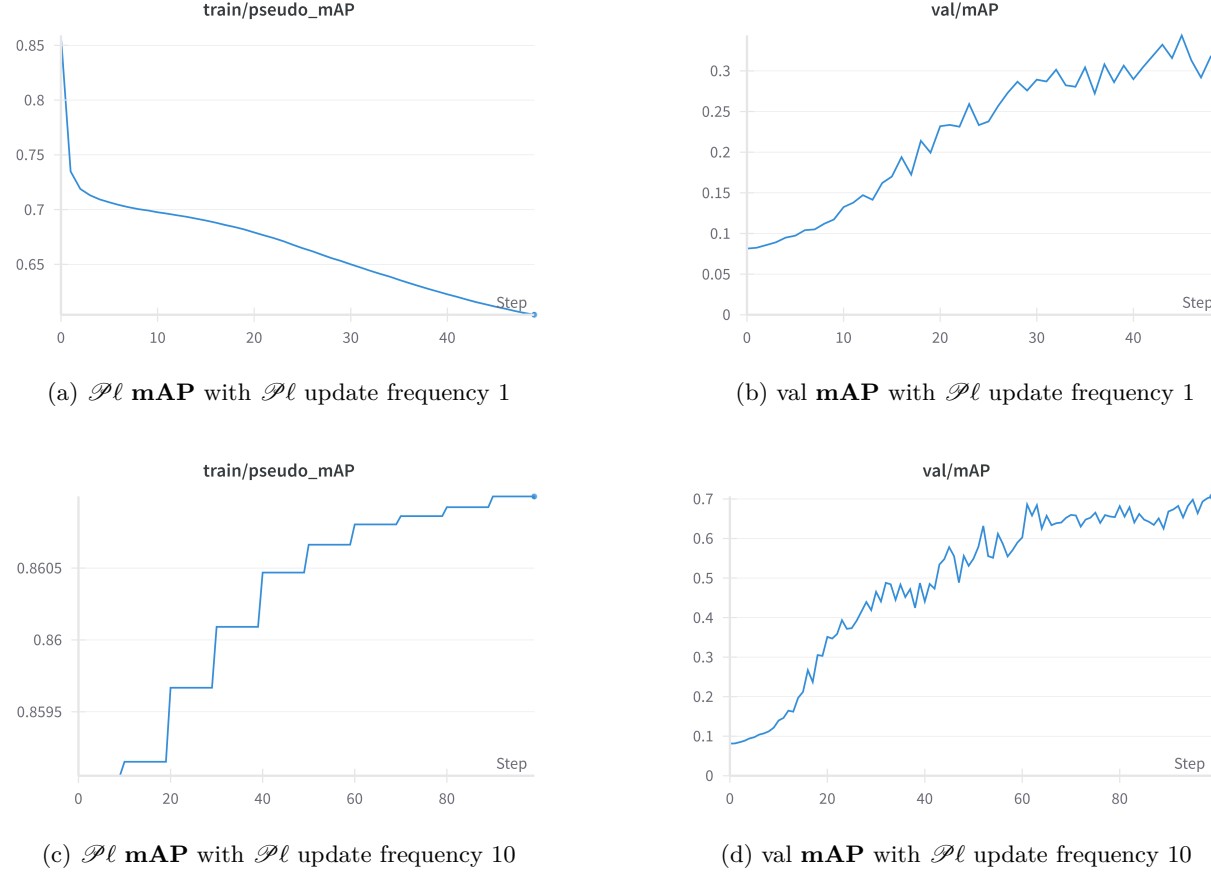

(a) $\mathscr{P}\ell$ **mAP** with $\mathscr{P}\ell$ update frequency 1

(b) val **mAP** with $\mathscr{P}\ell$ update frequency 1

(c) $\mathscr{P}\ell$ **mAP** with $\mathscr{P}\ell$ update frequency 10

(d) val **mAP** with $\mathscr{P}\ell$ update frequency 10

Figure 1: **mAP** values across epochs for different $\mathscr{P}\ell$ update frequencies

### 6.3 What was difficult

**Unavailability of a public codebase -** The main difficulty in verifying the central claims of the paper came from the unavailability of a public codebase.

**Computational restraints -** The process of generating the soft similarity aggregation vectors as part of the initialization step of training a classifier is highly time and compute intensive, as detailed out in section 4.6. The "Pre-Training Setting" subsection from section 4.1 of the original paper mentions that all unsupervised models are initialized and trained using the generated pseudo labels as initials for the unlabeled data. It would be highly insightful for the research community if the details on generating the pseudo labels such as optimizations, choice of using a $3 \times 3$ snippet size and computational requirements are made publicly available. Moreover, for large datasets such as MS-COCO, with around 82,081 images and NUSWIDE with around 150k images (in the trian splits), an open sourced cache of the pseudo label vectors would aid the research community in building further upon the work of the authors.

**Unavailability of pseudo-code -** We found it difficult to comprehend the following statements from section 3.2 of the original paper:

- "Once the pseudo labels are updated, we can fix them again and re-update the network parameters. This optimization procedure will continue until convergence occurs or the maximum number of epochs is reached" - This initial statement seems to update the pseudo labels and network parameters simultaneously in every epoch.

- "When the training is done, we fix the predicted labels and update the latent parameters of pseudo labels" - This statement could be interpreted to mean that only the network parameters are updated for certain epochs and then the pseudo labels are being updated.

A PyTorch style pseudo-code would have been highly insightful in understanding the optimization procedure. Nonetheless, we conducted experiments incorporating both these ideas, with the second approach being more superior as per our experiments.

**Unknown hyperparameters -** We were unable to find any reference for the value of the threshold parameter $\zeta$ as well as the value of $\sigma$ for the Gaussian distribution.

### 6.4   Communication with original authors

We tried contacting the authors over the official emails specified in the paper for clarifications related to the the above difficulties, but we did not get any response.

### 6.5   Future work

We plan to conduct experiments on subsets of other datasets and try to improve over the current optimization strategy.

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
