# OpenReview forum: "Reproducibility Study of "CDUL: CLIP-Driven Unsupervised Learning for Multi-Label Image Classification""
_TMLR — Rejected by TMLR_

### Review · Reviewer_NpGs · 2024-03-03

**Summary Of Contributions:**

Overall, this is a useful reproducibility study of the original paper which gives more details of experimental setting. The paper has made improvements addressing the original article's shortcomings in producibility. In the absence of open-source code, the authors create a codebase to verify the claims of the original paper: the effectiveness of  "the aggregation of global and local alignments" and "the gradient-aligment training method".

**Model description**

Just as the original paper, in the initialization stage, CLIP model with ResNet-50 is used as the image encoder and a fixed prompt is used to encode the text. While in training and inferencing stage, ResNet-101 is used for classifying.

**Hyperparameters**

The authors consider 3 x 3 snippet size as original paper is too small to find meaningful object and use sizes of [64, 32, 16]. What's more, the authors report that the threshold parameter is not specified in original paper and assign a constant of 0.5 to it. The authors have taken into consideration certain parameters, such as the epoch frequency for pseudo label updates and the value of *σ* in the Gaussian distribution, details that were not addressed in the original paper.

**Code**

The authors implement using PyTorch as deep learning framework. The codebase is configuration driven with the help of Hydra and Makefile.

**Result**

Through experimental results and comparison, the authors are unable to verify the effectiveness of claim 1 due to the time intensive computation when snippet size is 3 x 3. Also the cliam 2 cannot be completely reproduced using the original hyperparameters.

**Computational requirements**

The authors employ Big O notation to express the relationship between computational complexity and factors such as data volume, image size, and snippet size. Additionally, authors approximate time for cache generation for different snippet sizes.

**Audience:**

Yes

**Broader Impact Concerns:**

None.

**Claims And Evidence:**

Yes

**Requested Changes:**

Please refer to the weaknesses.

**Strengths And Weaknesses:**

#### Strengths

1. It is commendable that the authors have provided a reproducible and well-commented codebase with well structured configuration files. Furthermore, the authors supply the generated cache, encompassing both global and aggregate vectors, to enhance the reproducibility of the experiment.

#### Weaknesses

1. The overlap between section 1 and section 7 is noticeable, particularly in section 1.4 and 7.2, 1.5 and 7.3, 1.6 and 7.4. Addressing this redundancy would contribute to enhancing the overall conciseness of the paper.
2. There is a need for a more insightful exploration of the original experiments. A more extensive and thorough set of experiments in more datasets would likely strengthen the robustness of the claims made in authors' study. Maybe authors can provide additional rationale behind the appearance of "decreasing the pseudo label update frequency while increasing the quality of pseudo labels" .
3. To substantiate the argument proposed by authors, certain unknown hyperparameters like threshold parameter *ζ* as well as the value of *σ* for the Gaussian distribution need to be analyzed by choosing different values.

---

> ### Author Response · Authors · 2024-05-02
>
> > The overlap between section 1 and section 7 is noticeable, particularly in section 1.4 and 7.2, 1.5 and 7.3, 1.6 and 7.4. Addressing this redundancy would contribute to enhancing the overall conciseness of the paper.
>   >
>
>   We have updated the PDF with better conciseness and have kept the discussion to section 6.
>
>   > There is a need for a more insightful exploration of the original experiments. A more extensive and thorough set of experiments in more datasets would likely strengthen the robustness of the claims made in authors' study. Maybe authors can provide additional rationale behind the appearance of "decreasing the pseudo label update frequency while increasing the quality of pseudo labels" .
>   >
>
>   Decreasing the pseudo label update frequency would enable to provide a better gradient update as per equation (7) of the original paper. By decreasing the update frequency, we would reliably move in a particular direction based on the calculated confidence [equation (8) in original], instead of haphazardly updating the pseudo-labels in every epoch. We tried to validate this hypothesis empirically and found that this does indeed help. We have shown the results in Table 3 of our report.
>
>   > To substantiate the argument proposed by authors, certain unknown hyperparameters like threshold parameter *ζ* as well as the value of *σ* for the Gaussian distribution need to be analyzed by choosing different values.
>   >
>
>   We re-conducted the experiments by changing the values of the threshold parameter ζ  and found no change in the $mAP$ values of the initial pseudo labels generated using CLIP for any of the snippet sizes.

---

### Review · Reviewer_Hb68 · 2024-03-09

**Summary Of Contributions:**

For the problem of poor reproducibility caused by the lack of public code resources in a paper in ICCV 2023, an experimental study is proposed to verify the real effectiveness of the method proposed in the original paper. This paper mainly verifies two key parts of the method proposed in the original paper, i.e., global-local aggregation module and gradient-alignment training. The experimental results on the PASCAL VOC 2012 dataset show that the superiority of the method proposed in the original paper is difficult to achieve when some parameters are unknown.

**Audience:**

Yes

**Broader Impact Concerns:**

None.

**Claims And Evidence:**

Yes

**Requested Changes:**

Please refer to the "Weaknesses" part and the "Suggestions for Rebuttal" part.

**Strengths And Weaknesses:**

Strengths:
1. This work shows that the reproducibility of the previous paper in ICCV 2023 with existing information is contrary to the superiority claimed by the authors in the original paper. This means there may be some "tricks".
2. While this work is currently problematic, it serves as an example that may provide a good case for the CV community to promote fairness and openness of experimental and code resources.
3. This work reveals the impact of the snippet size on the results, which I appreciate very much after reading the original paper, since the choice of the snippet size is an important factor affecting the performance of the model, but the original paper does not provide any explanation for the setting of the 3 $\times$ 3 snippet size.


In order to objectively evaluate the work, I have read the original paper "CDUL: CLIP-Driven Unsupervised Learning for Multi-Label Image Classification". I do not believe that the current version of this work is suitable for publication, please refer to the "Weaknesses" part. If the authors can address my concerns, I would be willing to recommend acceptance, please refer to the "Suggestions for Rebuttal" part.

Weaknesses:
1. Since the contribution of the original paper is limited, which extends the CLIP model from single-label classification to multi-label classification, so its reproducibility study (the current version) is not enough to be published as a paper. However, the conclusions of this reproducibility work are somewhat important.
2. The readability of this paper is poor, and the content of this paper is not self-contained. The readers need to read the original paper to understand the content of this work.
3. In order to better support the conclusions of this paper, experiments need to be conducted on more datasets to show that the conclusion "the superiority of CDUL method is difficult to achieve when some parameter settings are unknown" is not accidental, since "No Free Lunch Theorem".

Suggestions for Rebuttal
1. It is necessary to add a section about the introduction of the original method to improve readability, that is, to describe the CDUL method in the most concise way possible.
2. It is necessary to show experimental results on more datasets, which should at least include the datasets appearing in the original paper and a few other commonly used datasets.
3. The results on the 3 $\times$ 3 snippet size should be reproduced for comparison with the original paper, and at least the results on the snippet size smaller than 16 $\times$ 16 should be shown to verify the conclusions in this paper.
4. Last but not least, some experiments, discussions and analyses of the methods involved in the original model are very important. These mainly include three aspects. First, the selection of the similarity score, the original paper uses simple cosine similarity. Second, the selection of the aggregation strategy, the original paper uses the min-max method and simple average operation. Third, the selection of the nonlinear transformation in the update method of the latent parameters of pseudo-labels, the original paper uses a simple Gaussian distribution. Different choices about these methods will significantly affect the final performance of the model. These are essentially model selection issues, and the study on these aspects will significantly enhance the contribution of this paper. It is obviously very challenging to solve the above problems, but at least some discussions and analyses about these problems are necessary. If the authors can provide some analyses of experimental results on the above problems, I will strongly appreciate and recommend this work.

---

> ### Author Response · Authors · 2024-05-02
>
> > It is necessary to add a section about the introduction of the original method to improve readability, that is, to describe the CDUL method in the most concise way possible.
>   >
>
>   We have updated the paper with a better explanation to the original CDUL method.
>
>   > It is necessary to show experimental results on more datasets, which should at least include the datasets appearing in the original paper and a few other commonly used datasets.
>   >
>
>   We have already mentioned some issues related to reproducing on other datasets - the number of images in case of MS-COCO (roughly 80k) and NUS-WIDE (roughly 150k) are fairly larger than PASCAL VOC 2012. We even tried considering [MS-COCO-mini](https://github.com/giddyyupp/coco-minitrain), but this also contains 25k images (which is still far greater than 5k from PASCAL VOC 2012). Since Generating the initial pseudo labels itself was highly compute intensive in terms of time (as shown in table 1), we only considered experimenting on PASCAL VOC 2012.
>
>   > The results on the 3 × 3 snippet size should be reproduced for comparison with the original paper, and at least the results on the snippet size smaller than 16 × 16 should be shown to verify the conclusions in this paper.
>   >
>
>   We have put up the experiment for generating the initial pseudo labels using the 3 x 3 snippet size. It has been running for more than a month now. We might have to wait for a week or two more. We will make the updates once we have the results.
>
>   > Last but not least, some experiments, discussions and analyses of the methods involved in the original model are very important. These mainly include three aspects. First, the selection of the similarity score, the original paper uses simple cosine similarity. Second, the selection of the aggregation strategy, the original paper uses the min-max method and simple average operation. Third, the selection of the nonlinear transformation in the update method of the latent parameters of pseudo-labels, the original paper uses a simple Gaussian distribution. Different choices about these methods will significantly affect the final performance of the model. These are essentially model selection issues, and the study on these aspects will significantly enhance the contribution of this paper. It is obviously very challenging to solve the above problems, but at least some discussions and analyses about these problems are necessary. If the authors can provide some analyses of experimental results on the above problems, I will strongly appreciate and recommend this work.
>   >
>
>   We have already tried to show the effect of the snippet size on the quality of the initial pseudo labels (Table 2) and have shown that decreasing the snippet size does have a positive effect but comes at a computation cost. This is a significant study which adds more insight to the direct selection of a 3 x 3 snippet size as made by the original paper. As for the update method, we do show a study which talks about the pseudo label update frequency rather than updating the labels in every iteration. We also reconducted some experiments on changing the threshold parameter ζ, where we found no change in the $mAP$ values of the initial pseudo labels generated using CLIP for any of the snippet sizes.

---

### Review · Reviewer_MLRM · 2024-03-18

**Summary Of Contributions:**

This paper studies the reproducibility of a published paper CDUL. As the report's authors mentioned, due to limited computational resources, they tried to use alternative settings to generate the pseudo labels to study the effectiveness of the aggregator. They employed ResNet-101 as the alignment network on Pascal 2012 to train it and update the pseudo labels during the training, validating its effectiveness.

**Audience:**

No

**Broader Impact Concerns:**

This work does not contain any broader impact concerns.

**Claims And Evidence:**

No

**Requested Changes:**

Please refer to the weakness above.

**Strengths And Weaknesses:**

Strengths:
Reproduce one of the previously published papers, CDUL.

Weakness:
In the Discussion section, the report’s authors mentioned that they were unable to verify claim 1 of the original paper due to the time-intensive computations and resource limitations associated with applying 3x3 settings. Instead, they opted to study alternative settings, such as 64x64, 32x32, and 16x16, to circumvent these challenges.

This raises several concerns:

If the current authors had access to additional resources, would they be able to reproduce the original study? We can not dismiss the original claim solely based on the limitations of the authors' resources. Moreover, can we accurately evaluate the original paper if the report's authors were unable to utilize the original paper's settings and instead employed alternative settings (64x64, 32x32, and 16x16)?

For clarification, we can not dismiss the credibility and effectiveness of the ChatGPT model (or any of the new LLM models) solely due to resource constraints or the computational intensity required to train these models from scratch. Although neither the original paper nor the report explicitly states that CDUL is a LLM, the analogy still implies the need for substantial computational resources to replicate the results, as mentioned in the report.

As reported in Table 2 and Section 7.1, the authors mentioned that they observed an increase in mAP in the 16x16 setting compared to the 64x64 setting. This observation confirms that the splitting approach with smaller snippets may have a positive impact on generating pseudo labels (higher mAPs) compared to larger splitting settings. Consequently, it prompts us to consider what conclusions could be drawn if the report’s authors had access to the appropriate resources and were able to apply the original paper's settings.

---

> ### Author Response · Authors · 2024-05-02
>
> > In the Discussion section, the report’s authors mentioned that they were unable to verify claim 1 of the original paper due to the time-intensive computations and resource limitations associated with applying 3x3 settings. Instead, they opted to study alternative settings, such as 64x64, 32x32, and 16x16, to circumvent these challenges.
>   >
>   > This raises several concerns:
>   >
>   > If the current authors had access to additional resources, would they be able to reproduce the original study? We can not dismiss the original claim solely based on the limitations of the authors' resources. Moreover, can we accurately evaluate the original paper if the report's authors were unable to utilize the original paper's settings and instead employed alternative settings (64x64, 32x32, and 16x16)?
>   >
>
>   We believe that a reproducibility study is a lot more than just exactly verifying the claims made by a paper. Our aim is to provide a ground for open discussion on the design choices used by the original method, and to provide an open source implementation of their method to the best of our understanding for the benefit of the larger research community.
>
>   > For clarification, we can not dismiss the credibility and effectiveness of the ChatGPT model (or any of the new LLM models) solely due to resource constraints or the computational intensity required to train these models from scratch. Although neither the original paper nor the report explicitly states that CDUL is a LLM, the analogy still implies the need for substantial computational resources to replicate the results, as mentioned in the report.
>   >
>
>   We would like to highlight that in the past as well, people have tried to conduct reproducibility studies as part of the [ML Reproducibility Challenge](https://reproml.org/), where the motive was not to just verify or refute the original paper, but was to provide more insights. Several previous accepted studies even show experiments conducted on publicly available cloud resources with very less compute and on a very constrained settings.
>
>   > As reported in Table 2 and Section 7.1, the authors mentioned that they observed an increase in mAP in the 16x16 setting compared to the 64x64 setting. This observation confirms that the splitting approach with smaller snippets may have a positive impact on generating pseudo labels (higher mAPs) compared to larger splitting settings. Consequently, it prompts us to consider what conclusions could be drawn if the report’s authors had access to the appropriate resources and were able to apply the original paper's settings.
>   >
>
>   We agree with the observation that the $mAP$ increases when the snippet size decreases, but the increase is extremely meagre compared to the cost of computation. We have put up the experiment to generate for the 3 x 3 size so that exact comparisons can be made with the original paper. We sincerely request a few more weeks to update the results.

---

> > ### Comment · Reviewer_MLRM · 2024-05-16
> >
> > Thanks for updating some details to the report. However, after reviewing your response, there's a need for clarification regarding how the image was divided. Did the division occur on a pixel-level, or a patch-level?  If pixel-level division was used, I recommend considering patch-level division for your posted Table's quality and reported time calculations.

---

> > > ### Author Response · Authors · 2024-05-24
> > > **Experiments using patch-level division**
> > >
> > > We have further experimented with different thresholds and it seems that setting the threshold parameter to 0 gives the best results. We have considered the snippet size to be 3 x 3 taking its meaning to be 3 x 3 = 9 number of patches (crops) of the original image. The compute time is much more manageable. These are the results on different thresholds for the PASCAL VOC 2012 dataset.
> > >
> > >
> > > | snippet size | threshold | final_lambda | mAP       |
> > > | ------------ | --------- | ------------ | --------- |
> > > | 3 x 3        | 0         | 0            | **89.30** |
> > > | 3 x 3        | 0         | 1            | **88.21** |
> > > | 3 x 3        | 0.05      | 0            | **89.31** |
> > > | 3 x 3        | 0.05      | 1            | **88.21** |
> > > | 3 x 3        | 0.1       | 0            | 83.88     |
> > > | 3 x 3        | 0.1       | 1            | 85.89     |
> > > | 3 x 3        | 0.2       | 0            | 83.88     |
> > > | 3 x 3        | 0.2       | 1            | 85.89     |
> > > | 3 x 3        | 0.3       | 0            | 83.88     |
> > > | 3 x 3        | 0.3       | 1            | 85.89     |
> > > | 3 x 3        | 0.4       | 0            | 83.88     |
> > > | 3 x 3        | 0.4       | 1            | 85.89     |
> > > | 3 x 3        | 0.5       | 0            | 83.88     |
> > > | 3 x 3        | 0.5       | 1            | 85.89     |
> > >
> > > Here `final_lambda` suggests if there was any averaging done according to:
> > > $$
> > > S^{\text {final }}=\frac{1}{2}\left(S^{\text {global }}+S^{\text {aggregate }}+\lambda\left|S^{\text {global }}-S^{\text {aggregate }}\right|\right)
> > > $$
> > >
> > > With this, we are certain that the strategy is able to produce better **mAP** than the global similarity vectors.
> > >
> > > Kindly allow us some time to produce results on some other datasets.

---

> ### Author Response · Authors · 2024-05-17
>
> The division is done by pixel level. We tried considering patch level division, but this concept didn't make sense to us since the original paper claims to be using a CLIP model with a ResNet vision encoder. We tried checking the codebase of CLIP for the ResNet case and were unable to find any `patch_size` argument in that case.
> The huggingface version of [CLIP](https://huggingface.co/docs/transformers/en/model_doc/clip#transformers.CLIPVisionConfig) has `patch_size` as an argument, but it is supported only for `openai/vit` variants. they haven't added the ResNet variants.
> Moreover, **figure 3 (a)** from the original paper has shown the division of the original image into snippets (unlabeled image crops). They do not show any features or "patches" that we think in terms of ViT.
>
> Edit: If the meaning of patch-level division for size 3 x 3 is taken as dividing the entire image in 3 x 3, with a total of 9 crops, then we re-ran the experiment. The **mAP** turns out to be 83.88 % for the initial pseudo labels. So whatever be the meaning of `snippet_size`, the **mAP** turns out to be lower than the simple global, being **85.9**.

---

### Author Response · Authors · 2024-05-15
**mAP (in %) of the final pseudo label vectors for a snippet size of 3 x 3**

We have finally been able to generate the aggregate vectors for a snippet size of 3 x 3. This required some multi-gpu inferencing modifications. But this still took a significant time for 5717 images of the training split of PASCAL VOC 2012 dataset. We summarize the results obtained on different snippet sizes.

| Snippet Size   | Ours   | Table 3 of the original paper  |
|----------------|--------|---------------------------------|
| Global         | **85.9** | 85.3 |
| $64 \times 64$ | 84.62  | (Not computed) |
| $32 \times 32$ | 84.99  | (Not computed) |
| $16 \times 16$ | 85.19  | (Not computed) |
| $3 \times 3$   | 85.47  | 90.3 |

Our experiments suggest that the global similarity vectors have a higher **mAP** over the "global local alignment strategy" for any of the snippet sizes ranging from a small 3 x 3 size to moderate 64 x 64.
The results have been updated in our submission and the clip cache has been added to our supplementary.

---

### Decision · Action_Editor_siH6 · 2024-05-25

**Recommendation:** Reject

**Comment:**

All three reviewers were in favor of reject. Reviewers' comments were not addressed well -- in particular, they thought that more investigation was needed to reach a final conclusion, and more insightful exploration was needed. Further ablation study is needed to show the affect of the hyperparmaters. Thus, the paper is rejected because it does not yet provide useful insight and actionable lessons for this method.

MLRM comments after discussion:
- After reviewing the recent response and based on the overall reviewer's feedback, the authors need to investigate more directions and conduct more experiments to reach a final conclusion.  For example, assuming an image with 224x224,  ** With 3x3 pixel-level splitting, each crop contains  9 pixels in total, which means finding an object per crop is not expected. The reported mAP is 85.47%, as listed in Table 1 of the report. ** With 3x3 patch-level splitting, each crop contains around 5476 pixels in total which that finding an object per crop is expected. The reported mAP is 83.88% in the last post. I think the intuition of leading these results is not clear.

NpGs comments after discussion:
- The overlap between section 1 and section 7 is noticeable, particularly in section 1.4 and 7.2, 1.5 and 7.3, 1.6 and 7.4. Addressing this redundancy would contribute to enhancing the overall conciseness of the paper.
- There is a need for a more insightful exploration of the original experiments. A more extensive and thorough set of experiments in more datasets would likely strengthen the robustness of the claims made in the authors' study. Maybe authors can provide the additional rationale behind the appearance of "decreasing the pseudo label update frequency while increasing the quality of pseudo labels".
- To substantiate the argument proposed by authors, certain unknown hyperparameters like threshold parameter ζ as well as the value of σ for the Gaussian distribution need to be analyzed by choosing different values.
- There is no response to my concerns.

Hb68 comments after discussion:
- The authors responded to my review 20 days after I submitted my official recommendation. I carefully read their responses, but sadly the authors did not address my concerns. Their responses provided too little useful information to support my attempt to re-evaluate this paper. I will maintain my original recommendation. The main reason is that the contribution of the original paper in ICCV 2023 is limited, and this reproducibility work lacks sufficient experimental analysis to support their conclusions. Furthermore, this paper does not inherently provide new insights and contributions.

**Audience:**

No, the study has limited insight for ML researchers.

**Claims And Evidence:**

Claims are not fully supported, and more experiments are needed.

**Resubmission Of Major Revision:**

The authors may consider submitting a major revision at a later time.